# Exploring the Loci Responsible for Awn Development in Rice through Comparative Analysis of All AA Genome Species

**DOI:** 10.3390/plants10040725

**Published:** 2021-04-08

**Authors:** Kanako Bessho-Uehara, Yoshiyuki Yamagata, Tomonori Takashi, Takashi Makino, Hideshi Yasui, Atsushi Yoshimura, Motoyuki Ashikari

**Affiliations:** 1Bioscience and Biotechnology Center, Nagoya University, Furo-cho, Chikusa, Nagoya, Aichi 464-8601, Japan; kanako.bessho.b3@tohoku.ac.jp; 2Graduate School of Life Sciences, Tohoku University, Aoba-ku, Sendai, Miyagi 980-8578, Japan; tamakino@tohoku.ac.jp; 3Faculty of Agriculture, Kyushu University, 744 Motooka, Nishi-ku, Fukuoka 819-0395, Japan; yoshiyuk@agr.kyushu-u.ac.jp (Y.Y.); hyasui@agr.kyushu-u.ac.jp (H.Y.); ayoshi@agr.kyushu-u.ac.jp (A.Y.); 4STAY GREEN Co., Ltd., 2-1-5 Kazusa-Kamatari, Kisarazu, Chiba 292-0818, Japan; ttakashi@staygreen.co.jp

**Keywords:** AA genome, awn, chromosome segment substitution lines, rice, wild species

## Abstract

Wild rice species have long awns at their seed tips, but this trait has been lost through rice domestication. Awn loss mitigates harvest and seed storage; further, awnlessness increases the grain number and, subsequently, improves grain yield in Asian cultivated rice, highlighting the contribution of the loss of awn to modern rice agriculture. Therefore, identifying the genes regulating awn development would facilitate the elucidation of a part of the domestication process in rice and increase our understanding of the complex mechanism in awn morphogenesis. To identify the novel loci regulating awn development and understand the conservation of genes in other wild rice relatives belonging to the AA genome group, we analyzed the chromosome segment substitution lines (CSSL). In this study, we compared a number of CSSL sets derived by crossing wild rice species in the AA genome group with the cultivated species *Oryza sativa* ssp. *japonica*. Two loci on chromosomes 7 and 11 were newly discovered to be responsible for awn development. We also found wild relatives that were used as donor parents of the CSSLs carrying the functional alleles responsible for awn elongation, *REGULATOR OF AWN ELONGATION 1* (*RAE1*) and *RAE2*. To understand the conserveness of *RAE1* and *RAE2* in wild rice relatives, we analyzed *RAE1* and *RAE2* sequences of 175 accessions among diverse AA genome species retrieved from the sequence read archive (SRA) database. Comparative sequence analysis demonstrated that most wild rice AA genome species maintained functional *RAE1* and *RAE2*, whereas most Asian rice cultivars have lost either or both functions. In addition, some different loss-of-function alleles of *RAE1* and *RAE2* were found in Asian cultivated species. These findings suggest that different combinations of dysfunctional alleles of *RAE1* and *RAE2* were selected after the speciation of *O*. *sativa*, and that two-step loss of function in *RAE1* and *RAE2* contributed to awnlessness in Asian cultivated rice.

## 1. Introduction

Rice is a major staple food that provides the caloric requirements for nearly one-fourth of the world’s population [1]. Two cultivated rice species, *Oryza sativa* and *O. glaberrima*, were independently domesticated in Asia and Africa from wild progenitors, *O. rufipogon* and *O. barthii*, respectively [2,3]. Compared with the wild progenitors, the cultivated rice species share a common set of morphological characteristics, including non-shattering seeds, white pericarp color, erect tiller growth, and short-awned or awnless seed [4,5,6,7]. These domesticated traits contributed to increasing rice yield, grain quality, and cultivation efficiency.

Wild rice species develop an awn, which is a long extension of the lemma tip. Under natural conditions, awns provide seed protection from predators and a unique means of seed dispersal via attachment to human clothes or animal fur [8]. However, in agriculture, awns represent a disadvantage as these needle-like structures cause skin irritation to farmers during harvesting and threshing. Rice awns also reduce bulk density, resulting in loose packing during storage. Unlike awns in barley or wheat, which contribute to grain filling, rice awns are not photosynthetically active because they lack chlorenchyma [9,10]. Some studies have reported that awn loss increases rice grain number, such that rice awns have a negative impact on yield [11,12]. The genes responsible for awn development have been suggested to have pleiotropic effects on grain number and morphology. Identifying the genes conditioning awn will provide insight into the domestication process in rice and may lead to increased rice grain production by breeding.

The long awn of wild progenitors was eliminated through rice domestication. Several major genes associated with awn development have been identified, including *An-1/RAE1* [11,13], *LABA1/An-2* [12,14], *DL* and *OsETT2* [15], and *RAE2/GAD1/GLA* [16,17,18]. Genes suppressing awn formation have also been identified, including the YABBY transcription factor, *TOB1* [19], and *GLA1,* which encodes the mitogen-activated protein kinase phosphatase [20]. In addition, a total of 35 loci with major and minor effects on rice awning in *O*. *sativa* have been reported in the Gramene database (https://archive.gramene.org/, accessed on 5 April 2021).

*REGULATOR OF AWN ELONGATION 1* (*RAE1*, *LOC_Os04g28280*) [13] and *RAE2* (*LOC_Os08g37890*) [16] were mainly targeted for the selection of an awnless phenotype in Asian rice domestication. *RAE1* encodes the bHLH transcription factor and was also reported as *An-1* [11]. *O*. *sativa* ssp. *japonica* carries a 4.4 kb transposable element (TE) in the promoter region of *RAE1* to decrease its expression level, whereas *O*. *sativa* ssp. *indica* has a single nucleotide polymorphism (SNP) in the second exon, which causes an early translational stop [11,13]. *RAE2* encodes *Epidermal Patterning Factor-Like protein 1* (*EPFL1*), which acts as a small secretary peptide in panicles [16] and has also been reported as *GAD1* [17] and *GLA* [18]. Several SNPs occur in the coding region of *RAE2* in *O*. *sativa*, which cause frameshifts that change the number of cysteines essential for creating disulfide bonds that lead to suitable protein conformations. Dysfunctional alleles of *RAE1* and *RAE2* have been selected during Asian rice domestication [11,16].

To identify novel loci regulating awn development and to determine the degree of conservation of *RAE1* and *RAE2* genes in other wild rice relatives belonging to the AA genome group (Appendix A) [21], we analyzed chromosome segment substitution lines (CSSL) and the whole-genome sequence data of a set of wild and cultivated rice species from the sequence read archive (SRA) database. CSSLs represent the whole genome of a donor line (i.e., wild rice species) in small and contiguous chromosome segments in a background (recurrent) line [22]. CSSLs allow the detection of responsible loci distributed across the genome using fewer plants than other approaches such as quantitative trait locus (QTL) analysis, which use segregating populations (i.e., F_2_) or recombinant inbred lines [23,24,25,26]. To date, many series of CSSLs have been developed to identify complex trait loci in rice between cultivated species and their wild relatives [27,28,29,30,31]. In addition, the accumulation of genome information for many plant species by next-generation sequencing (NGS) techniques impacts rice research. Whole-genome sequencing has been performed on model plants as well as many wild plant species, and their sequence information was collected in the National Center for Biotechnology Information (NCBI) database. These sequence data can be used for comparative analysis of target genes in many species and accelerate the research understanding the gene selection through plant domestication.

In the present study, we evaluate 11 CSSLs for identifying novel loci regulating awn development, which includes one CSSL named RRESL that is developed in this study. All the CSSLs had been fixed those recurrent parents into *O*. *sativa* ssp. *japonica*, and it makes us allow to compare the effect of locus from donor parent by comparison of target trait and genotype. We also compared the *RAE1* and *RAE2* genes of the donor parents of each CSSL to examine the relationship between causal mutations and the awn phenotype. Genome sequence data from 175 wild and cultivated rice accessions were analyzed for the sequence comparison of *RAE1* and *RAE2* and revealed the conserveness and selection history of both genes in the AA genome group.

## 2. Results

### 2.1. Development of a CSSL Whose Donor Parent Is O. glumaepatula and Recurrent Parent Is Koshihikari (O. sativa ssp. japonica)

The effects of chromosome segments derived from a donor parent can be widely compared by fixing the genetic background of recurrent parents with those of closely related species. We collected 10 sets of CSSL from previous studies whose recurrent parent is fixed with Asian cultivated rice, *O*. *sativa* ssp. *japonica* crossed with wild and cultivated rice species of the AA genome group (Table 1).

These 10 sets of CSSL included five CSSLs with the recurrent parent Taichung 65 (T65), which is *O*. *sativa* ssp. *japonica,* and donor parents are *O*. *rufipogon* [32], *O*. *nivara* [32], *O*. *glumaepatula* [33], and two sets of *O*. *meridionalis* [34]. The recurrent parent of the other five CSSLs is Koshihikari, which is also *O*. *sativa* ssp. *Japonica,* and donor parents are Kasalath (*O*. *sativa*) [24], *O*. *nivara* [35], *O*. *rufipogon* [30], *O*. *glaberrima* [29], and *O*. *barthii* [31]. Two CSSL sets (MER-IL and MER-IL[MER]) were derived from crossing T65 and W1625 (*O*. *meridionalis*); their recurrent parents are T65 but have different cytoplasmic DNA. MER-IL has T65 cytoplasmic DNA, and MER-IL[MER] has W1625 cytoplasmic DNA.

In addition to these 10 CSSLs from previous studies, we developed a new CSSL named RRESL whose donor parent is *O*. *glumaepatula*, a South American wild rice and whose recurrent parent is Koshihikari (Figure 1A). To develop RRESL, we used Koshihikari and IRGC105666 (*O*. *glumaepatula*). After producing F_1_, we produced BC_4_F_2_, BC_5_F_2,_ and BC_6_F_2_ by successive backcross with Koshihikari. Marker-assisted selection (MAS) was performed to select the lines carrying one chromosome segment from the donor parent and covering the entire genome by all lines. Finally, RRESL was composed of 35 individuals of chromosome-substituted lines (Figure 1B) (see details in Materials and Methods).

### 2.2. Awn Phenotype in Wild Rice Species, Cultivated Rice Species, and CSSLs

According to the morphological data obtained from Oryzabase (https://shigen.nig.ac.jp/rice/oryzabase/, accessed on 15 July 2018) and previous studies [11,16,36], all the wild rice species possess awns. We presented 3 wild rice species: *O*. *nivara*, *O*. *meridionalis*, and *O*. *glumaepatula*, which are wild species producing long awns (Figure 2A). On the other hand, African and Asian cultivated rice do not possess awns, whereas *O*. *sativa* ssp. *indica* (*aus*) Kasalath produces awns (Figure 2B). Four lines were selected from different CSSLs (GLU-IL112, MER-IL113, KKSL210, BSL14), possessing representative awn phenotype (Figure 2C). Comparing the awn length among wild cultivars and selected CSSL lines, the awn lengths of CSSLs were consistently shorter than that of the donor parent (Figure 2D).

To identify the loci responsible for awn development, we planted 11 sets of CSSL in the research field. We measured the ratio of the awned spikelet in one panicle and the awn length per CSSL according to the standard evaluation system (SES) of rice provided by the International Rice Research Institute (IRRI) [37] during a period of 2 to 3 years (Appendix A). Data collected in 2016 are shown in Figure 3 as an example. Two to six lines in each CSSL showed the awned phenotype (Figure 3A–E and Appendix A). The awned spikelet ratio per panicle varied from 7 to 100%. Each CSSL showed diverse awn lengths, and the lines whose recurrent parents were T65 and Koshihikari had maximum awn lengths of 2.0 (Figure 3A,B) and 6.0 cm (Figure 3C–E), respectively. This suggests awn lengths tend to be shorter in T65 recurrent parent.

### 2.3. Identification of the Responsible Loci for Awn Development

We mapped the chromosome regions responsible for awn development on 12 rice chromosomes by comparing the genotypes and awn phenotypes of each CSSL (Figure 4, see detail in Material and Method). Chromosome-substituted regions are referred from the graphical genotypes previously reported (listed in Appendix A). Most CSSLs had functional regions for awn development on chromosomes 4 and 8. Notably, all lines carrying the chromosome 4 segment of the donor parent showed the awned phenotype throughout the study period (Appendix A), except MER-IL[MER]. To date, two genes responsible for awn development have been reported on chromosome 4: *An-1*/*RAE1* [11,13] (chr4: 16731738..16735336) and *LABA1/An-2* [12,14] (chr4: 25959399..25963504). All CSSLs carrying the *RAE1* region of the donor parents (WK1962-IL16, 17, WK56-IL10, GLU-IL112, MER-IL113, 114, KKSL210, NSL10, 11, RSL11, GLSL13, 14, BSL14, and RRESL14; data are shown in Figure 3A–E and Appendix A) showed the awned phenotype. MER-IL[MER]15, 16, and 17 carry the chromosome 4 segment of W1625 (*O*. *meridionalis*), but these fragments do not cover the *RAE1* region, and these 3 lines do not have an awn (Appendix A). Three CSSL lines carrying the chromosome segment harboring the *LABA1* location (NSL11, GLSL14, and RRESL14) showed the awned phenotype. Seven CSSL lines (WK56-IL22, MER-IL[MER]34, NSL18, GLSL25, 26, BSL29, and RRESL25) carrying the long arm region of chromosome 8 overlapping *RAE2* location (chr8: 23998787..24000176) showed a significant awned phenotype. Six CSSLs (GLU-IL115, MER-IL116 and 117, MER-IL[MER] 18 and 19, and BSL18; Figure 3 and Appendix A) carrying the short arm of chromosome 5 in each donor parent presented awn. This region overlapped with a locus reported as a QTL, *An7*, derived from *O*. *glumaepatula* [38], which suggests that *O*. *meridionalis* and *O*. *barthii* may also carry functional *An7*. Two loci on chromosome 1 for awn development have been reported as *An9* and *An10* in *O*. *meridionalis* [36]. MER-IL102 carrying the long chromosome 1 segment of *O*. *meridionaris* covered the *An9* and *An10* regions, and the chromosome segment of the donor parent of MER-IL[MER]2 overlapped with the *An9* locus, which indicates that the donor parent W1625 carries functional *An9* and/or *An10*. WK56-IL2 carried a segment of the long arm of the chromosome 1 region covering the *An10* locus, which suggests that WK56 (*O*. *nivara*) may carry functional *An10*. Recently, a locus responsible for awn length was detected on chromosome 2 and designated as *qAWL2* [39]. This locus was narrowed down between a pair of simple sequence repeat (SSR) markers, RM13335 and RM13349, within a 157.4-kb region. Since this chromosome region overlaps the region covered by WK1962-IL7, GLU-IL106, and KKSL206, the QTL in chromosome 2 in W1926, WK35, and Kasalath may correspond to *qAWL2*.

We newly detected two loci responsible for awn development on chromosomes 7 and 11 (Figure 4). The short arm region of chromosome 7 was overlapped among 3 lines of CSSL, GLU-IL121 (*O*. *glumaepatula*) and MER-IL[MER]27 and 28 (*O*. *meridionaris*), and it was narrowed down between a pair of SSR markers RM1353 and RM7121 within a 2.31-Mb region. The short arm region of chromosome 11 in WK56-IL29 (*O*. *nivara*) and GLU-IL130 (*O*. *glumaepatula*) was overlapped between a pair of SSR marker RM7557 and RM3625 within a 4.33-Mb region. To our knowledge, this is the first study to report the loci regulating awn development in these regions.

Among detected loci regulating the awn phenotype, *RAE1* and *RAE2* are well-conserved regions for awn development in wild rice species (Figure 4). Next, we further compared the sequences of *RAE1* and *RAE2* to identify the causal mutations using the Sanger sequencing data and deposited sequences of wild and cultivated rice species.

### 2.4. RAE1 and RAE2 Sequences in Donor and Recurrent Parents

The loss-of-function alleles of *RAE1* and *RAE2* were selected during Asian rice domestication [9,10,11,12]. Dysfunctional *RAE1* alleles with a 4.4-kb transposable element (TE) in the promoter region that causes decreased *RAE1* expression and a 1-bp deletion in the second exon that causes translational suspension by changing the stop codon have been reported in *japonica* and *indica* rice, respectively [11,13]. In *RAE2*, six cysteines are conserved for functional *RAE2* protein; however, several dysfunctional alleles have lost or gained extra cysteines in Asian cultivated rice due to a frameshift caused by an insertion or deletion of several base pairs in the second exon [16,17].

To test whether the donor and recurrent parent of CSSLs possessed functional *RAE1* and *RAE2*, we sequenced both genes in all CSSL parental lines using the Sanger sequencing method (Appendix A). A summary of the sequence results is shown in Table 2. T65 and Koshihikari, which were the recurrent parents of all CSSLs, both carried dysfunctional alleles of *rae1* and *rae2*. Since neither T65 nor Koshihikari exhibit awns, the awnless phenotype corresponds to both genotypes. By contrast, the African cultivated species *O*. *glaberrima* has retained both functional alleles of *RAE1* and *RAE2*. The awnless phenotype of *O*. *glaberrima* is caused not by *RAE1* and *RAE2* but by a loss of function in locus *RAE3* on chromosome 6, which has not yet been identified [13]. *O*. *sativa* ssp. *indica* (*aus*) Kasalath has retained a functional allele of *RAE1* and a dysfunctional allele of *rae2*, resulting in the awned phenotype. Similarly, W0106 (*O*. *rufipogon*) has a functional allele of *RAE1* and a dysfunctional allele of *rae2*. The CSSL lines harboring the *RAE1* region of chromosome 4 of Kasalath or W0106 (KKSL210 or RSL11) showed the awned phenotype, but those harboring the *RAE2* region of chromosome 8 (KKSL224 or RSL22) did not show awn (Appendix A). These results support the presence of functional *RAE1* but dysfunctional alleles of *rae2* in both Kasalath and W0106.

Other wild rice species, as the donor parents, possessed functional alleles of both *RAE1* and *RAE2*, and all the lines showed awn phenotype (Table 2).

### 2.5. Awn Development Is Regulated by Different Functional Combinations of RAE1 and RAE2 among the AA Genome Group

To understand the conservation of *RAE1* and *RAE2* function for the awned phenotype in wild and cultivated rice species, we compared the sequences of both genes using whole-genome sequence data from nine *Oryza* species of the AA genome group (*O*. *sativa* ssp. *japonica*, *O*. *sativa* ssp. *indica*, *O*. *rufipogon*, *O*. *nivara*, *O*. *glumaepatula*, *O*. *meridionalis*, *O*. *glaberrima*, *O*. *barthii*, and *O*. *longistaminata*). Sequencing reads from 175 accessions were mapped to the reference genome of *O. sativa* ssp. *japonica* and SNPs called. To judge the individuals with or without a TE insertion in the promoter region of *RAE1*, we focused on the boundary region of targeted TE insertion. Detection of the target region covering the 200 bp of transposon edge and flanking region of the TE insertion boundary (Appendix A) allowed us to detect the specific TE in the *RAE1* promoter region distinguished from the same sequence of TE in other genomic location. In the case of *O*. *meridionalis*, short reads of samples were not mapped to the TE inserted region upstream of *RAE1* of *O*. *sativa* ssp. *japonica* genome. This is because there were multiple indels in the *O*. *meridionalis* genome near the TE inserted region compared with the *O*. *sativa* ssp. *japonica* genome (Appendix A). Therefore, we mapped short reads of 11 *O*. *meridionalis* samples from the SRA database to the *O*. *meridionalis* genome ver. 1.3 instead of using the *O*. *sativa* ssp. *japonica* genome as a reference. We confirmed that the short reads from *O. meridionalis* were mapped on the *O. meridionalis* genome seamlessly (Appendix A); thus, it suggests that there was no TE insertion in the *RAE1* promoter region.

First, we retrieved 2.5 kb of the *RAE1* gene coding region and the region around the TE insertion at the *RAE1* promoter in 57 accessions of the Asian wild rice species *O*. *rufipogon* and 33 Asian cultivated species, including 21 *japonica* and 12 *indica* varieties. Comparative sequence analysis detected 33 common SNPs and seven insertions and deletions (indels) in the *RAE1* coding region and one 4.4-kb TE in the *RAE1* promoter region. According to those common variants, we identified two major mutations in Asian wild and cultivated rice varieties (Appendix A) and divided *RAE1* into four haplotypes (*RAE1*-hap1 to *RAE1*-hap4) (Figure 5A). *RAE1*-hap1, which had no TE insertion in the promoter region and no 1-bp deletion in the second exon, was conserved in most *O*. *rufipogon* and may be a functional allele of *RAE1*. *RAE1*-hap2 had no 4.4-kb TE insertion but had a 1-bp deletion in the second exon of *RAE1*, which was found in *O*. *sativa* ssp. *indica* and may be a dysfunctional allele of *rae1*. *RAE1*-hap3 had 4.4-kb TE conserved in most *O*. *sativa* ssp. *japonica* and may be a dysfunctional allele of *rae1*. *RAE1*-hap4 had both 4.4-kb TE and 1-bp deletion in the second exon of *RAE1*, which may be a dysfunctional allele of *rae1*. *RAE1*-hap1 was conserved in most wild rice species and *O*. *glaberrima*, and *RAE1*-hap2 and *RAE1*-hap3 were detected in *O*. *sativa* ssp. *indica* and ssp. *japonica*. *RAE1*-hap4 was not observed in any accessions (Appendix A).

Next, we retrieved a 1.4 kb *RAE2* sequence from each of the 175 accessions. We detected eight SNPs and 11 indels in this region. The *RAE2* gene encoded a peptide-protein containing six cysteine residues (6C) that are essential for suitable conformation leading to awn development. Mis-conformation mutants caused loss of *RAE2* function [16,17]. We identified five major indels in the second exon and classified six *RAE2* haplotypes (*RAE2*-hap1 to *RAE2*-hap6) using these mutations (Appendix A and Figure 5B). *RAE2*-hap1 had no indels that were conserved among most *O*. *rufipogon*; *RAE2*-hap2 had a 6-bp deletion compared to *RAE2* of *O*. *rufipogon*. These two haplotypes conserved 6C in the mature peptide region and, therefore, may be functional alleles. A 1-bp deletion in *RAE2*-hap3 occurred only in *O*. *sativa* ssp. *japonica*, where it caused a premature stop codon causing truncation that destroyed the fifth and sixth cysteine residues in RAE2, resulting in 4C type RAE2, which may be a dysfunctional allele of *rae2*. *RAE2*-hap4 (2-bp deletion) and hap5 (2-bp deletion) create a RAE2 protein frameshift to become a 7C mature peptide. The rare haplotype *RAE2*-hap6 (1-bp insertion) occurred only in *O*. *sativa* ssp. *japonica* and made a RAE2 protein frameshift to become a 7C mature peptide. RAE2 (7C type) may be a dysfunctional allele of *rae2* (Figure 5B).

We categorized 175 rice accessions (123 wild and 52 cultivated rice; 18 *O*. *barthii*, 19 *O*. *glaberrima*, 18 *O*. *glumaepatula*, 11 *O*. *meridionalis*, 16 *O*. *longistamianta*, three *O*. *nivara*, 57 *O*. *rufipogon*, 12 *O*. *sativa* ssp. *indica*, and 21 *O*. *sativa* ssp. *japonica*) into four groups according to the haplotype combination at *RAE1* and *RAE2* (Appendix A). Group I is a combination of *RAE1*-hap1 and *RAE2*-hap1 or *RAE2*-hap2 (*RAE2*-hap1/2) (RAE1: functional, RAE2: functional), group II is a combination of *RAE1*-hap1 and *RAE2*-hap3/4/5/6 (RAE1: functional, rae2: dysfunctional), group III is a combination of *RAE1*-hap2/3/4 and *RAE2*-hap1/2 (rae1: dysfunctional, RAE2: functional), and group IV is a combination of *RAE1*-hap2/3/4 and *RAE2*-hap3/4/5/6 (rae1: dysfunctional, rae2: dysfunctional). The global distribution of these four groups is shown in Figure 5C. All accessions of wild rice species used in this study, including *O*. *barthii*, *O*. *longistaminata*, *O*. *meridionalis*, *O*. *glumaepatura*, *O*. *nivara*, and *O*. *rufipogon*, were classified into group I, with few exceptions. One accession in *O*. *longistaminata* (6.3%) was classified into group II (*RAE1*-hap1 and *RAE2*-hap4). *RAE2*-hap4 is a unique haplotype observed only in this accession of *O*. *longistaminata*. Two accessions in *O*. *rufipogon* (3.5%) were classified into group II (*RAE1*-hap1 and *RAE2*-hap5) (Figure 5C, Appendix A). One accession in *O*. *rufipogon* (1.7%) was classified into group III (*RAE1*-hap3 and *RAE2*-hap1).

Interestingly, all *O*. *glaberrima* (awnless) African cultivated rice were classified into group I (*RAE1*-hap1 and *RAE2*-hap1/2) (Figure 5C and Appendix A), which may contain functional alleles of both genes. The wild rice *O*. *barhii* (awned), which is the ancestor of *O*. *glaberrima*, was also classified into group I. These results indicate that *RAE1* and *RAE2* were not selected for awnlessness in African rice domestication. By contrast, Asian cultivated rice species *O*. *sativa* ssp. *japonica* and *O*. *sativa* ssp. *indica* carried diverse combinations of the *RAE1* and *RAE2* haplotypes. In *O*. *sativa* ssp. *japonica,* one accession (4.7%) was classified into group I, one accession (4.7%) was classified into group II, four accessions (19.0%) were classified into group III, and 15 accessions (71.4%) were classified into group IV. However, in *O*. *sativa* ssp. *indica*, three accessions (25%) were classified into group I, three accessions (25%) were classified into group II, one accession (8.3%) was classified into group III, and five accessions (41.7%) were classified into group IV. This result indicates that >70% of *O*. *sativa* ssp. *japonica* possess dysfunctional alleles of a combination of *rae1* and *rae2*; and around half of *O*. *sativa* ssp. *indica* possess dysfunctional alleles of both genes.

## 3. Discussion

In rice, the awn is a conspicuous trait that is considered to have been influenced by domestication. To date, several genes have been identified for awn development in rice, including *An-1/RAE1* [11,13], *LABA1/An-2* [12,14], *RAE2/GAD1/GLA* [16,17,18], *TOB1* [19], and *GLA1* [20]; among these, *An-1/RAE1*, *LABA1/An-2*, and *RAE2/GAD1/GLA* appear to have been selected through Asian rice domestication [11,12,16]. To explore the novel loci for regulating awn development and to clarify the conservation of *RAE1* and *RAE2* gene function among the AA genome rice group, we examined 11 sets of CSSL by comparing genotypes and awn phenotypes.

According to the comparative analysis among 11 CSSLs, two to six lines in each CSSL showed the awn phenotype, and awn length was consistently shorter than that of the donor parent. This suggests that each wild rice species of the AA genome group possesses two to six genes that promote awn elongation, and a single locus was insufficient to attain the awn length of the donor parent. We detected several loci related to awn development on chromosomes 1, 2, 5, 7, and 11. Some of these loci were previously reported, as *An9* and *An10* on chromosome 1, *qAWL2* on chromosome 2, and *An7* on chromosome 5, although not all of these genes have been identified to date. Two regions on chromosomes 7 and 11 have not been reported for awn development; these are unidentified loci conserved in the donor parents of *O. glumaepatura*, *O*. *meridionalis,* and *O*. *nivara* in our CSSL set. Comparing multiple sets of CSSL made it possible to identify the loci that have not been detected by QTL analysis or single CSSL analysis.

We also detected that chromosome segments located on chromosomes 4 and 8 were conserved in most of all wild rice species. There are two genes reported on chromosome 4, *RAE1* and *LABA1*. All the CSSL lines harboring the *RAE1* locus showed awn phenotype, but a few lines harboring the *LABA1* locus present awn. These observations suggest that *RAE1*, not *LABA1*, is the major and common locus on chromosome 4 for awn development among the AA genome group. In addition, most of the CSSL lines harboring *RAE2* locus also showed awn. Sanger sequence results show the functional *RAE1* and *RAE2* correlate with awn phenotypes in CSSLs. Previous QTL studies of wild rice species also found that these two regions on chromosomes 4 and 8 have significant effects on awn length [40,41]. These results indicate that functional *RAE1* and *RAE2* are conserved in AA genome rice species.

The number of loci regulating awn development has been reported so far, but a comprehensive understanding about which wild species possess which loci has not been clarified. By using the CSSL set derived from all AA genome group species, we were able to estimate which species have which genes/loci in common and to find new loci. On the other hand, the study using CSSLs also has a weak point. CSSL cannot account for the effects of cooperative genes located in different chromosome regions. Indeed, none of the awned lines among the various CSSLs used in this study attained the awn length or awned ratio of the donor parents. Additive or synergetic effects by multiple loci would explain awn length and awned seed ratio in original wild species. Combination of the loci by the crossing between awned CSSL lines would reveal the relationship of responsible genes. Pyramiding all of the loci detected in this research would mimic awn length and ratio in the original wild donor parent.

*RAE1/An-1* influences the formation of awn primordia by regulating cell division [11], and *RAE2/GAD1/GLA* regulates awn length by promoting cell division at the tip of the lemma [16,17]. *An-1/RAE1* and *RAE2/GAD1/GLA* work independently for awn development and show additive effects [13,42]. The combination haplotypes of *RAE1* and *RAE2* in donor and recurrent parents are consistent with their awn phenotypes, except for *O*. *glaberrima*. It is reported that the awnless phenotype of *O. glaberrima* is caused not by *RAE1* and *RAE2* but by a loss of function in the *RAE3* locus on chromosome 6 [13]. Awn phenotype was not observed in all CSSL lines harboring the *RAE3* locus in this study. This suggests *RAE3* locus cannot produce awn itself, but the loss of function of *RAE3* diminished awn phenotype, even possessing functional *RAE1* and *RAE2*. So far, the *RAE3* was not identified, yet cloning of *RAE3* elucidates the molecular mechanism for awn formation by *RAE3* and helps to understand African rice domestication.

Since it is revealed that most donor parents of CSSLs carried functional *RAE1* and *RAE2* alleles by observation of CSSL and Sanger sequence, we performed a comparative analysis of the *RAE1* and *RAE2* sequence to assess the functional conservation of both genes in wild rice species. The *RAE1* and *RAE2* sequences analysis in 175 accessions of wild rice AA species obtained from the NCBI SRA database indicated that most of the accessions of wild rice species had functional *RAE1* and *RAE2*, except for a few accessions of *O*. *longistaminata* and *O*. *rufipogon*. *RAE2*-hap4, which is observed in one accession of *O*. *longistaminata,* is distinct from the haplotypes of *O*. *sativa*; therefore, this mutation in *RAE2*-hap4 would have coincidently occurred in the accession of *O*. *longistaminata*. The dysfunctional haplotypes, *RAE1*-hap3 and *RAE2*-hap5, found in a few accessions of *O*. *rufipogon,* were also detected in *O*. *sativa*. N6205, which is one of the *O. rufipogon* accessions, carries *RAE1*-hap3, which is conserved in most *japonica* accessions, and W1669 and W1715, which are *O. rufipogon* accessions, carry *RAE2*-hap5, which is conserved in most *indica* accessions. We hypothesize two possibilities. One is that *japonica* and *indica* were derived from different accessions of *O. rufipogon*, and the other is that it might reflect the admixture of *O. rufipogon* and *O*. *sativa* [43]. To clarify these possibilities, the population analysis using a larger accession panel is necessary.

This research suggests that *RAE1* and *RAE2* are the major loci for awn development in wild relatives in the AA genome rice group. By contrast, within subspecies of *O*. *sativa* ssp. *japonica* and *indica*, the selection patterns of *RAE1* and *RAE2* differed; with >70% of *O*. *sativa* ssp. *japonica* lines possessing loss-of-function alleles of both genes and around 40% of *O*. *sativa* ssp. *indica* lines lost both genes’ function (Figure 5C). These percentages were slightly different from those reported previously [16,18], perhaps because this study used a smaller number of individuals for sequence comparison using SRA data, which requires sufficient read depth to identify the TE region compared to conventional PCR analysis [11]. Mutation points and haplotype occupancy differed between *japonica* and *indica*. *RAE1*-hap2 and *RAE2*-hap5 can be observed only in *indica*, while *RAE1*-hap3 and *RAE2*-hap3/*RAE2*-hap6 can be observed only in *japonica*. This suggests that loss of function in *RAE1* and *RAE2* in each subspecies have occurred by independent mutations that might have occurred in different *O. rufipogon* accessions. The haplotype analysis of *RAE2/GLA* by Zhang et al. [18] showed that the speciation of a *RAE2/GLA* natural variant might occur prior to the divergence of *japonica* and *indica*, which is consistent with the results of our study. In *O. sativa*, some lines lost only *rae1,* and some lost only *rae2*, suggesting that there were two steps of loss of function for *RAE1* and *RAE2*. Further, each *RAE1* and *RAE2* can produce awns independently, and it is necessary to lose both genes’ function for awnless phenotype in Asian rice domestication. These findings indicate that mutations of *RAE1* and *RAE2* occurred after the speciation of *O*. *sativa*, and subsequently, different combinations of dysfunctional *RAE1* and *RAE2* alleles were selected in Asian rice domestication.

In this study, we compared a number of CSSL sets derived from AA genome species as donor parents and confirmed that each wild species possessed previously reported loci related to awn development; during this process, we also found new loci. This research re-realized us that multiple CSSLs are useful materials for genetic studies such as gene mapping and uncovering the selection histories of the target genes. In future studies, the combined application of CSSL analyses and publicly available NGS data will allow us to further elucidate the gene selection history of the AA genome rice group and identify new loci for targeted domestication traits.

## 4. Materials and Methods

### 4.1. Plant Materials

All CSSLs used in this study are listed in Table 1. Donor parents (provider of a particular chromosome segment of CSSL) and recurrent parents (genetic background of CSSL) are also listed in Table 1. All donor and recurrent parents belong to the AA genome group. CSSLs whose recurrent parent is *O*. *sativa* ssp. *japonica* cv. T65 include WK1962-IL (donor: WK1962 (*O. rufipogon*)), WK56-IL (donor: WK56 (*O. nivara*)), GLU-IL (donor: WK35 (*O. glumaepatula*)), MER-IL (donor: W1625 (*O. meridionalis*)), and MER-IL[MER] (donor: W1625 (*O. meridionalis*)). WK56 was originated from pure line isolation from IRGC105715. MER-IL[MER] and has the cytoplasm of W1625 (*O*. *meridionalis*), whereas MER-IL and other CSSLs have that of T65. Graphical genotypes of these 5 sets of CSSL and seeds were obtained from Oryzabase (https://shigen.nig.ac.jp/rice/oryzabase/, access on: 10 June 2019) and Kyushu University. CSSLs whose recurrent parent is *O*. *sativa* ssp. *japonica* cv. Koshihikari include KKSL (donor: Kasalath (*O. sativa* ssp. *indica*)), NSL (donor: W0054 (*O. nivara*)), RSL (donor: W0106 (*O. rufipogon*)), GLSL (donor: IRGC104038 (*O. glaberrima*)), BSL (donor: W0009 (*O. barthii*)), and RRESL (donor: IRGC105666 (*O. glumaepatula*)). These 6 CSSLs have the cytoplasm of Koshihikari. Graphical genotypes of these 6 sets of CSSL and seeds were obtained from Honda Research Institute and Nagoya University. We developed and firstly reported RRESL in this study. Other CSSLs are previously reported; WK1962-IL [32], WK56-IL [32], GLU-IL [33], MER-IL [34], MER-IL[MER] [34], KKSL [24], NSL [35], RSL [30], GLSL [29], and BSL [31].

### 4.2. Marker-Assisted Selection (MAS) for Developing RRESL

F_1_ plant derived from a cross between Koshihikari and IRGC105666 was backcrossed with Koshihikari to produce 78 BC_1_F_1_ plants. BC_1_ plants were then backcrossed with Koshihikari three or four times, without marker-assisted selection (MAS), to produce BC_4_F_1_ and BC_5_F_1_ generation. Whole-genome genotyping was performed in the 140 BC_4_F_1_ and 120 BC_5_F_1_ using SNP markers distributed across the 12 rice chromosomes for MAS. Fifteen BC_4_F_1_ and 15 BC_5_F_1_ were selected and performed self-pollination or backcrossing with Koshihikari. Among 15 BC_4_F_2_, 22 BC_5_F_2,_ and 5 BC_6_F_2_, the lines (11 BC_4_F_2_, 19 BC_5_F_2,_ and 5 BC_6_F_2_) having one to two long, contiguous chromosome segments of IRGC105666 in Koshihikari genetic background were selected for constituting RRESL. Genomic DNA was extracted from the leaf blade of the samples using the ISOPLANT method and eluted in distilled water. Marker-assisted selection (MAS) using 149 single nucleotide polymorphisms (SNPs) (Appendix A) using the AcycloPrime-FP Detection System and Fluorescence Polarization Analyzer (Perkin Elmer Life Science, Boston, MA, USA) following manufacture protocol was conducted. The SNP markers, which were developed using the Build 2 pseudomolecules of *O. sativa* ssp. *japonica* cv. Nipponbare, and evenly distributed across the 12 rice chromosomes at an average marker interval of 3.5 Mb. All backcrossed lines were cultivated at the experimental field of the Honda Research Institute in Kisarazu, Chiba, Japan, following a conservational agricultural method.

### 4.3. Determination of Substituted Segments in RRESL for Making Graphical Genotype

The lengths of substituted chromosome segments in RRESL were determined based on the SNP marker position (Appendix A). A chromosome segment flanked by two markers of donor parent type was considered homozygous of donor parent type; a chromosome segment flanked by two markers of recurrent parent type was considered homozygous of recurrent parent type; and a chromosome segment flanked by one marker of donor type and one marker of recurrent parent type was considered as recombination occurred between two markers.

### 4.4. Growth Conditions and Phenotypic Evaluation

Plant materials were grown in the field of Nagoya University at Togo, Aichi and in the field of Kyushu University at Kasuya, Fukuoka, Japan following the conventional agronomic calendar. We took matured seed pictures of some lines of donor parents, recurrent parents, and CSSLs with awn as representations. Ten seeds were used for awn length measurement and shown in Figure 2D.

For awn phenotypic evaluation of 11 sets of CSSL, we used 10 plants per CSSL to measure the awned spikelet ratio per panicle and awn length. The percentage of awned spikelets was determined from awned spikelets divided by all seeds on the main stem panicle, where the awned phenotype was defined as awns > 3 mm in length on mature grains. The average awn length was measured on the apical spikelet of each primary branch on the main stem panicle. We measured the awn phenotype over 2 to 3 years because the awn is a labile phenotype that depends on the environment. Phenotypic data for most CSSLs were collected for 3 years (2011, 2014, and 2016, or 2015, 2016, and 2017); however, data of WK1962-IL and WK56-IL were collected in 2014 and 2016, and those of BSL and RRESL were collected in 2016 and 2017. Data collected in years other than 2016 followed the standard evaluation system (SES), as defined by IRRI [37] (Appendix A). Quantitative data for 2016 are shown as representative data in Figure 3 and Appendix A.

### 4.5. Chromosomal Localization of Responsible Loci for Awn Development

Chromosome regions harboring loci responsible for awn development were decided based on the phenotypic observation of awns more than twice within the 2-to-3-year examination period. Awn-related chromosome regions are highlighted by different colors depends on CSSL in Figure 4 based on the graphical genotype of each CSSL (provided in the original paper of each CSSL [24,29,30,31,32,33,34,35] and Figure 1B) and positional information of the substituted chromosome with markers were listed in Appendix A. For example, awn phenotype was observed two times in WK1962-IL16, which is carrying 0.17–18.5 Mb of chromosome 4 segment of WK1962 (*O. rufipogon*), and in WK1962-IL17, which is carrying 18.5–20.1 Mb of chromosome 4 segment of WK1962 (Appendix A). These results and information were converted to illustrations in Figure 4. We highlighted 0.17–20.1 Mb of chromosome 4 segment by yellow indicating WK1962-IL. Positional information of the genes and loci responsible for awn development were obtained from previous marker-based studies: *An9* and *An10* [36], *An-1/RAE1* [11,13], *LABA1/An-2* [14], *An7* [38], *RAE2/GAD1* [16], and *RAE3* [13].

### 4.6. Identifying the Functional Mutations of RAE1 and RAE2 in the Donor and Recurrent Parents of CSSL

We extracted genomic DNA from the leaf blade of the donor and recurrent parents of CSSLs using the ISOPLANT method and amplified the TE region of *RAE1* and the coding sequence region of *RAE1* and *RAE2*. DNA amplification was performed using PCR with the following conditions: 95 °C for 5 min; 30 cycles of 94 °C for 30 s, 60 °C for 30 s, and 72 °C for 2 min; and a final cycle of 72 °C for 5 min. Reactions were carried out in 96-well PCR plates in 25-μL volumes containing 1 μmol/L of each primer, 200 μmol/L of dNTPs, 5 ng of DNA template, 2 mmol/L MgCl_2_, 2.5 μL 10× buffer, and 1 U of the Prime Star HS (Takara Bio, Tokyo, Japan) as a PCR reagent. PCR products were subjected to electrophoresis using 1% agar/TAE buffer and gel purified using Wizard SV Gel and the PCR Clean-Up System (Promega, Madison, USA) for the Sanger sequencing using a capillary sequencer (ABI 3130 Genetic Analyzer, Applied Biosystems, Foster city, USA. Translation and multiple sequence alignments were performed using the MUSCLE method in Geneious software version 9.0. The primer set is listed in Appendix A. The sequence alignments are in Appendix A.

### 4.7. Evaluation of Functional RAE1 and RAE2 Allele in AA Genome Rice Species Using SRA Data

Whole-genome sequence data for 18 *O*. *barthii*, 19 *O*. *glaberrima*, 18 *O*. *glumaepatula*, 11 *O*. *meridionalis*, 16 *O*. *longistamianta*, three *O*. *nivara*, 57 *O*. *rufipogon*, 12 *O*. *sativa* ssp. *indica*, and 21 *O*. *sativa* ssp. *japonica* individuals (Appendix A) were obtained from the NCBI SRA database (http://www.ncbi.nlm.nih.gov/sra, accessed on 5 April 2021) and DNA Data Bank of Japan (DDBJ) (https://www.ddbj.nig.ac.jp/dra/index.html, accessed on 5 April 2021) and converted to the fastq format. Low-quality reads were removed using the fastp ver. 0.20.0 program [44]. Sequence reads for each individual were aligned to the *O*. *sativa* ssp. *japonica* genome IRGSP-1.0 (downloaded from Ensembl plants database on 26 September 2019) using the BWA ver. 0.7.12-r1039 aligner [45]. Redundant reads were excluded using the Samtools ver. 1.9 program [46], and genomic realignment was conducted using the ABRA2 ver. 2-2.22 software [47] for precise indel identification. The variant call was performed by mpileup with default settings. Mapping of short reads of *O. meridionalis* to the *O. meridionalis* genome v1.3 was performed following the method above.

To identify a reported TE insertion [11] in the promoter region of *RAE1* as suggested in *O*. *sativa* ssp. *japonica* (chr4:16735371..16739771), we focused on the genomic boundaries of the inserted regions chr4:16735371 and chr4:16739771. Because the same sequence of the target TE is present in other locations in the genome of *O. sativa japonica* and other AA genome species, multi-mapped reads into the rice genome were excluded from the analysis. Detection of the target region covering the 200 bp before and after the target TE insertion boundary was performed using the bedtools ver. 2.29.1 program [48] (Appendix A). TE insertion was judged at 80% coverage based on the breadth of sequence alignment within the TE (>160/200 bp) and outside of the TE (>160/200 bp), as shown in Appendix A. No TE insertion was identified at less than 20% coverage within the TE (<40/200 bp) and more than 80% coverage outside of the TE (>160/200 bp), as shown in Appendix A.

### 4.8. Alignment of TE Inserted Region among AA Genome Species

Genome sequence data of AA genome species and *O. punctata,* which is belonged to BB genome species as an outgroup, were retrieved from the Ensembl plant database. Sequences around the TE inserted region upstream of *RAE1* (chr4:16735371..16739771) for each individual were obtained and aligned using the MUSCLE method in Geneious software version 9.0. Since *O. sativa* ssp. *japonica* sequence with TE (4.4 kb) cannot be aligned with others, NNNNNNN sequence was inserted instead of TE.

## Figures and Tables

**Figure 1 plants-10-00725-f001:**
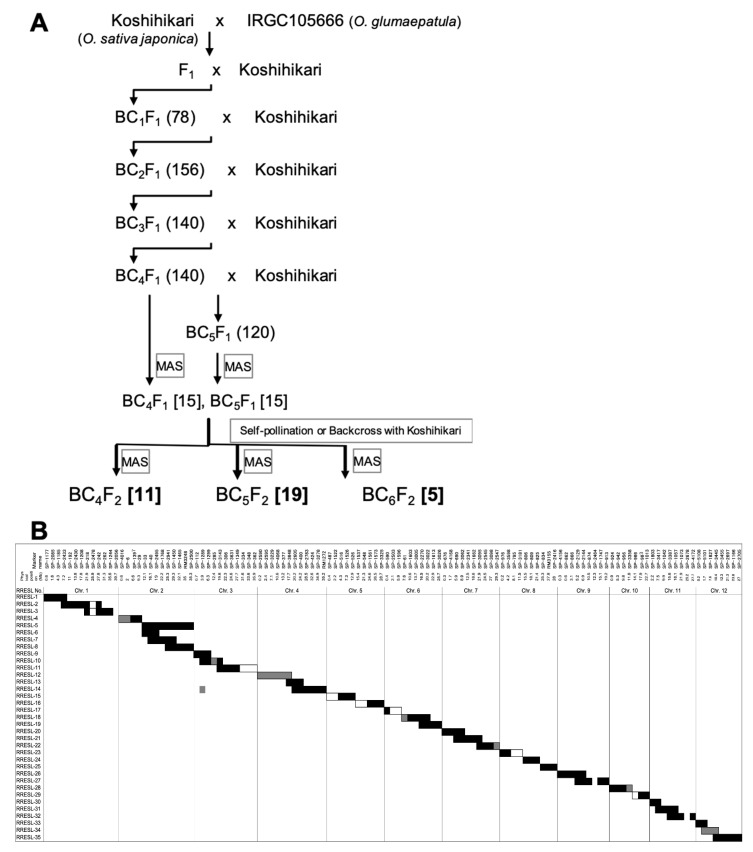
Flowchart of the development of RRESL. (**A**) Breeding scheme for developing RRESL carrying IRGC105666 (*O*. *glumaepatula*) chromosome segments in the Koshihikari (*O*. *sativa* ssp. *japonica*) genetic background. Numbers in round and square brackets indicate the numbers of lines produced in each backcross generation and candidate lines for RRESL selected by marker-assisted selection (MAS), respectively. Bold numbers in brackets were finally selected from the resulting chromosome segment substitution lines (CSSL). (**B**) Graphical representation of the genotypes of the 35 lines in RRESL. White and black bars indicate homozygous chromosomal segments derived from Koshihikari and IRGC105666, respectively. Gray bars represent heterozygous regions. Single nucleotide polymorphism (SNP) markers used for MAS are indicated above the table with their physical positions (Mb) for each chromosome (see also Appendix A).

**Figure 2 plants-10-00725-f002:**
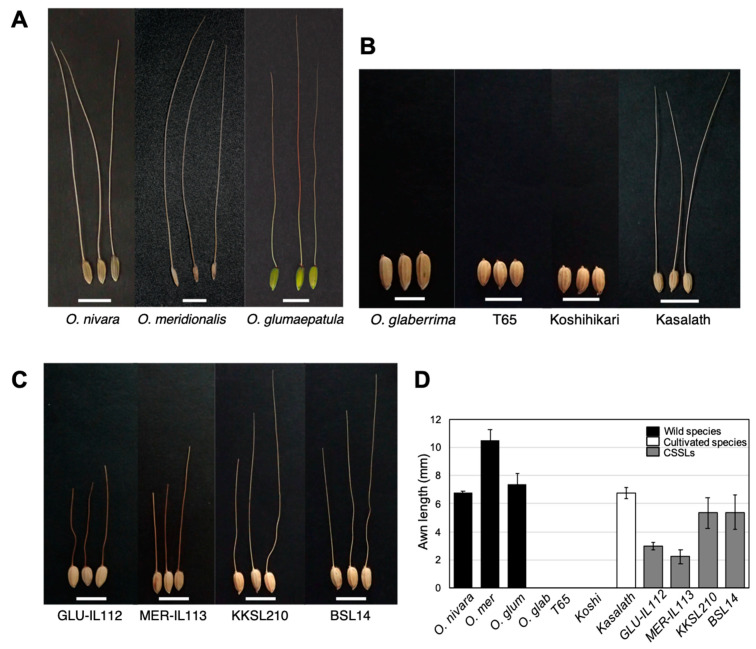
Awn phenotypes of materials. (**A**,**B**) Seeds of selected CSSL parents among (**A**) wild rice species and (**B**) cultivated rice species. African cultivated rice: *O. glaberrima*, Asian cultivated rice: T65 (*O. sativa, japonica*), Koshihikari (*O. sativa, japonica*), Kasalth (*O. sativa, indica* (*aus*)). (**C**) Seeds of several CSSLs possessing awns. (**D**) Awn lengths for each line. Average awn length of ten seeds±SD. Scale bar = 1 cm. O.mer, *O. meridionalis*; O.glum, *O. glumaepatula*; O.glab, *O. glaberrima*; T65, Taichung 65; Koshi, Koshihikari.

**Figure 3 plants-10-00725-f003:**
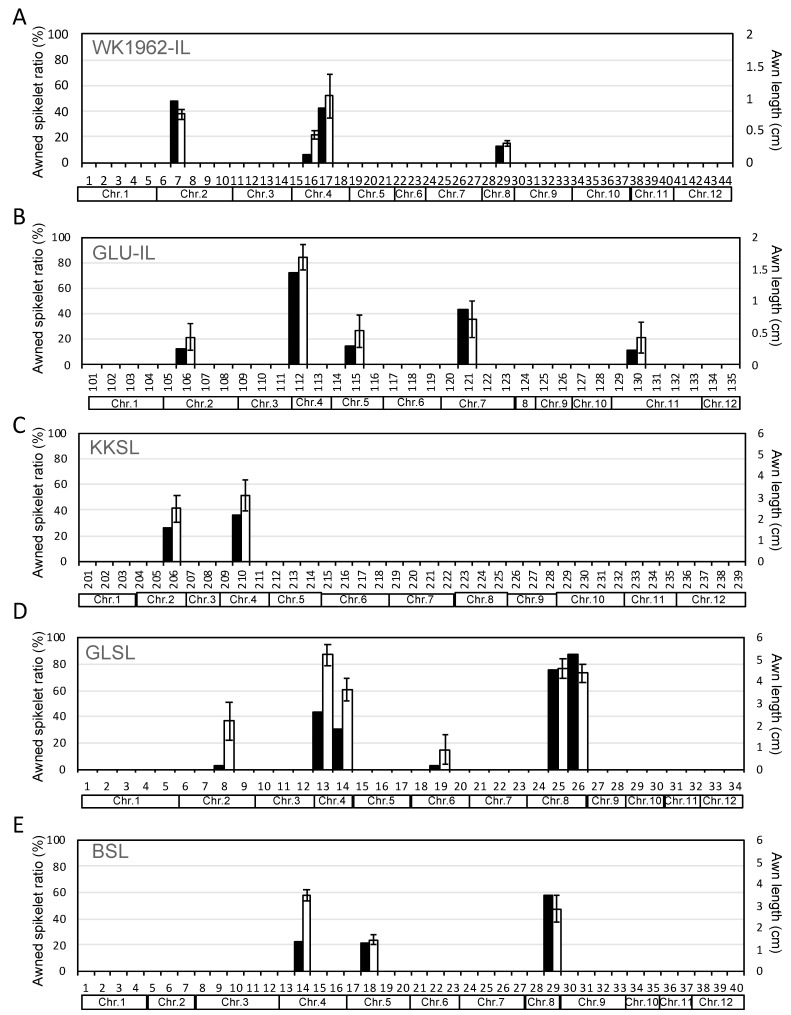
Comparison of awn traits among CSSLs. Awned spikelet ratio per panicle and awn length among CSSLs whose recurrent parent was T65: (**A**) WK1962-IL and (**B**) GLU-IL. (**C**–**E**) Awned spikelet ratio per panicle and awn length among CSSLs whose recurrent parent was Koshihikari: (**C**) KKSL, (**D**) GLSL, and (**E**) BSL. Awn phenotype data obtained in 2016 was presented. Awn phenotype data in other CSSLs are presented in Appendix A. Black bars indicate awned spikelet ratio per panicle; white bars indicate awn length. Numbers above the chromosome boxes indicate line number constituting each CSSL. Awn length data are means ± SD.

**Figure 4 plants-10-00725-f004:**
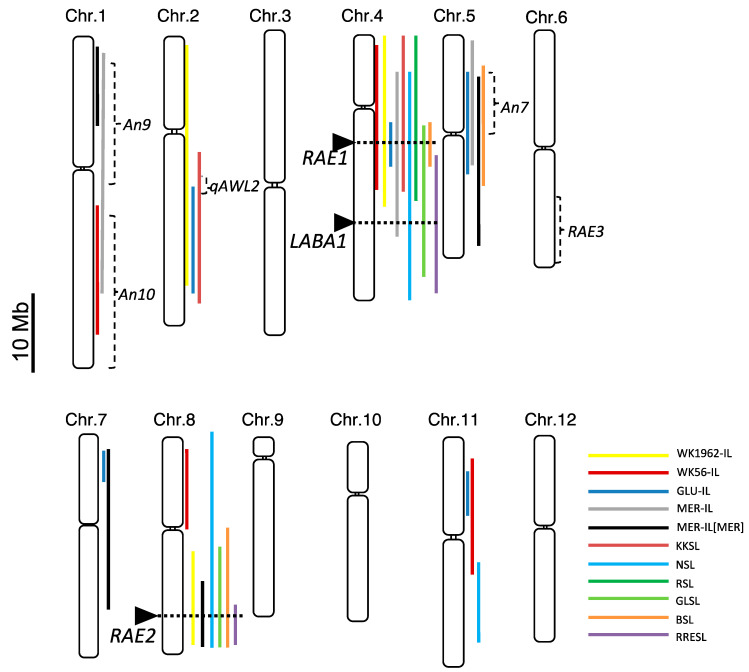
The responsible loci for awn development in each donor parent. White boxes indicate 12 rice chromosomes and the small box between big white boxes indicate centromere. Colored bars indicate the substituted chromosome region of the donor parent from each CSSL carrying the loci responsible for awn development. Black arrowheads indicate the positions of the genes *RAE1*, *LABA1*, and *RAE2*. Dotted brackets indicate the loci of *qAWL2*, *An7*, *An9*, *An10*, and *RAE3* previously identified through quantitative trait loci (QTL) or linkage mapping analysis.

**Figure 5 plants-10-00725-f005:**
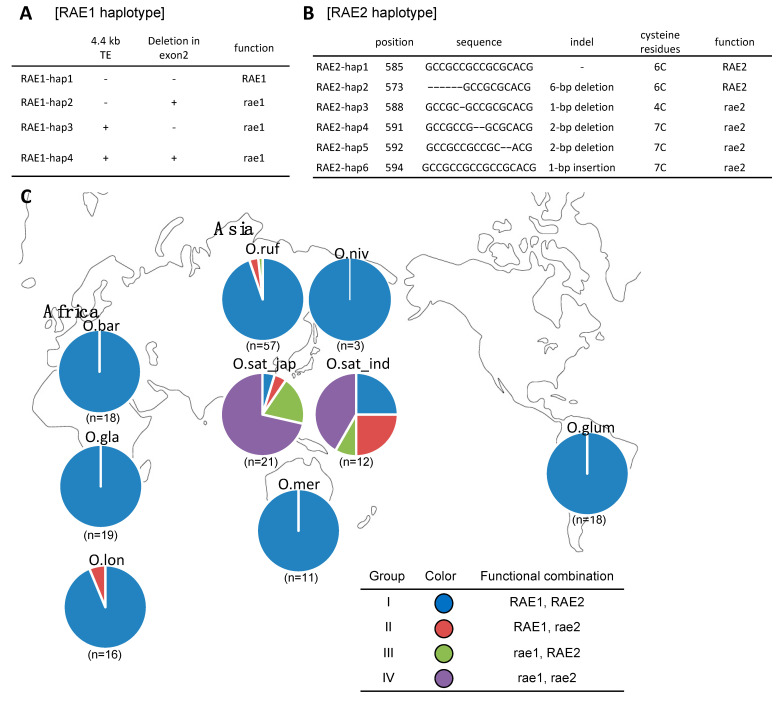
Frequency of the combination of *RAE*1 and *RAE*2 haplotypes. (**A**,**B**) Haplotypes in *RAE1* (**A**) and *RAE2* (**B**) categorized based on the comparison between *O. rufipogon* and *O. sativa* ssp. *japonica/indica*. (**C**) Combinations of the *RAE1* and *RAE2* haplotypes among the AA genome species were classified into four groups. The pie chart indicates the percentage of each group, where blue, red, green, and purple indicate groups I, II, III, and IV, respectively. Numbers below the pie charts are the numbers of accessions used for this analysis. Green and orange squares indicate cultivated rice species and their wild progenitors in Africa and Asia, respectively. O.bar, *O. barthii*; O.gla, *O. glaberrima*; O.lon, *O. longistaminata*; O.mer, *O. meridionalis*; O.glum, *O. glumaepatula*; O.ruf, *O. rufipogon*; O.niv, *O. nivara*; O.sat_jap, *O. sativa* ssp. *japonica*; O.sat_ind, *O. sativa* ssp. *indica*.

**Table 1 plants-10-00725-t001:** The list of CSSLs used in this study.

CSSL Name	Line Number	Donor Parent	Recurrent Parent	Reference
WK1962-IL	44	WK1962 (*O. rufipogon*)	T65 (*O. sativa japonica*)	[32]
WK56-IL	34	WK56 (*O. nivara*)		[32]
GLU-IL	35	WK35 (*O. glumaepatula*)		[33]
MER-IL	36	W1625 (*O. meridionalis*)		[34]
MER-IL [MER]	42	W1625 (*O. meridionalis*)		[34]
KKSL	39	Kasalath (*O. sativa*)^a^	Koshihikari (*O. sativa japonica*)	[24]
NSL	26	W0054 (*O. nivara*)		[35]
RSL	33	W0106 (*O. rufipogon*)		[30]
GLSL	34	IRGC104038 (*O. glaberrima*) ^a^		[29]
BSL	40	W0009 (*O. barthii*)		[31]
RRESL	35	IRGC105666 (*O. glumaepatula*)		this study

^a^ Two donor parents, Kasalath (*O. sativa*) and IRGC104038 (*O. glaberrima*), are Asian and African cultivated species, respectively. Other donor parents are wild relatives in AA genome group.

**Table 2 plants-10-00725-t002:** Sequence comparison of *RAE1* and *RAE2* in recurrent and donor parent.

Cultivated/Wild	Line Name	*RAE1* TE	*RAE1* 2nd_exon_G	*RAE1* Function	*RAE2* Cys no.	*RAE2* Function	Awn Phenotype
cultivated	Koshihikari (*O. sativa japonica*)	+	G	*rae1*	4	*rae2*	-
cultivated	T65 (*O. sativa japonica*)	+	G	*rae1*	4	*rae2*	-
cultivated	IRGC104038 (*O. glaberrima*)	-	G	*RAE1*	6	*RAE2*	-
cultivated	Kasalath (*O. sativa indica, aus*)	-	G	*RAE1*	7	*rae2*	+
wild	IRGC105715 (*O. nivara*)	-	G	*RAE1*	6	*RAE2*	+
wild	W0054 (*O. nivara*)	-	G	*RAE1*	6	*RAE2*	+
wild	W1962 (*O. rufipogon*)	-	G	*RAE1*	6	*RAE2*	+
wild	W0106 (*O. rufipogon*)	-	G	*RAE1*	7	*rae2*	+
wild	W1625 (*O. meridionalis*)	-	G	*RAE1*	6	*RAE2*	+
wild	WK35 (*O. glumaepatula*)	-	G	*RAE1*	6	*RAE2*	+
wild	IRGC105666 (*O. glumaepatula*)	-	G	*RAE1*	6	*RAE2*	+
wild	W0009 (*O. barthii*)	-	G	*RAE1*	6	*RAE2*	+

In the *RAE1*_TE column, “+” indicates that it has 4.4 kb TE in the promoter region of *RAE1*, “-” indicates no TE. In the *RAE1*_2nd_exon_G column, “G” represented it is same as a reference allele of *O. rufipogon* at G780 position. There are no lines carrying deletion as suggested in *indica* variety for loss of function of *An-1* (*RAE1*) [11]. In the awn phenotype column, “-” indicates awnless, and “+” indicates awned phenotype. Gray color indicates dysfunctional allele of *RAE1* and *RAE2* gene and awnless phenotype. The awnless phenotype of *O. glaberrima* is caused not by *RAE1* and *RAE2*, but by loss-of-function in locus *RAE3.*

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
