# Peer review of "Exploring the Loci Responsible for Awn Development in Rice through Comparative Analysis of All AA Genome Species"

_plants, 2021, doi:10.3390/plants10040725_

Round 1

Reviewer 1 Report

This work explores the the loci responsible for awn development in rice. Manuscript is well written and structured and therefore it can be accepted for publication after a careful spell chek.

Author Response

We appreciate your review of our manuscript and your comment about the English spell check. We carefully checked the spelling and revised it as needed.

Reviewer 2 Report

I would like to start my review from the Materials and methods chapter, which is unfortunately very poorly prepared. Materials should be properly described because it is not clear which of these materials were obtained earlier and which were derived in this work. Only from the title, it is known that species with the AA genome were used for crossing.

Growth conditions should be included in the phenotyping subchapter.

We do not know much about the process of the chromosomal location of the awn-forming loci, but there are many results regarding the chromosomal localization in the text.

The section on the identification of functional RAE1 and 2 mutations is properly described.

I am not sure if all the information contained in the paragraph about functional analysis based on SRA data is necessary. Half of them should be moved to the Results chapter.
The abstract - could be shorter - too much introductory information, but the fragment I marked in the introduction - defining the goal of the research - should be included in it, because the purpose of the research was not precisely defined in the abstract.
The chapter presenting the results should follow the same order as the methodology, so a poor methodology deprives us of the possibility of parallel tracking and verifying whether the presented results were obtained correctly.

In the results, I have marked as an example one of the fragments about MAS that lack the methodology. A lot of results lack methodology either.

Moreover, in the results chapter, there are very often discussion elements, so it should either be separated or combined into one chapter the Results and discussion.

What has been called a discussion is a summary actually.

Pictures of O. glumaepatula, MER-IL113 and Kasalath are blurred.

I am also not sure if all supplementary materials should be supplementary or be a part of the main article. Please, reconsider this once again.

Summing up, the results presented in the paper are very good and should be published, but the form in which they were presented is unacceptable. Therefore, I propose to publish this manuscript after a thorough revision. I would like to emphasize that these are technical corrections - concerning the text, not merit. For the revision of this manuscript, please start by writing a comprehensive methodology. 

Best regards,

Author Response

Comments and Suggestions for Authors

I would like to start my review from the Materials and methods chapter, which is unfortunately very poorly prepared. Materials should be properly described because it is not clear which of these materials were obtained earlier and which were derived in this work. Only from the title, it is known that species with the AA genome were used for crossing.

Growth conditions should be included in the phenotyping subchapter.

<Author’s response>

Thank you for your comments. According to your suggestion, we added the detailed information about the materials obtained in the earlier work and established in this work at the materials and method section. We also described the growth condition and phenotypic evaluation at “4.1 Plant materials” and “4.4. Growth conditions and phenotypic evaluation” in materials and methods.  

We do not know much about the process of the chromosomal location of the awn-forming loci, but there are many results regarding the chromosomal localization in the text.

<Author’s response> 

Thank you for your comments. We described how we indicated the chromosome location of the awn-forming loci. In the revised manuscript, this information was added in the Materials and Methods part “4.5. Chromosomal localization of responsible loci for awn development”.

The section on the identification of functional RAE1 and 2 mutations is properly described.

I am not sure if all the information contained in the paragraph about functional analysis based on SRA data is necessary. Half of them should be moved to the Results chapter.

<Author’s response>

Thank you for your comments and suggestion. We modified the section on the identification of functional RAE1 and RAE2 mutations with more information. Some sentences which describe the functional analysis based on SRA data were moved to the results chapter.

The abstract - could be shorter - too much introductory information, but the fragment I marked in the introduction - defining the goal of the research - should be included in it, because the purpose of the research was not precisely defined in the abstract.

<Author’s response>  

Thank you for your comments and suggestion. We now shortened the abstract and we revised to describe and define the goal of our research.

The chapter presenting the results should follow the same order as the methodology, so a poor methodology deprives us of the possibility of parallel tracking and verifying whether the presented results were obtained correctly.

In the results, I have marked as an example one of the fragments about MAS that lack the methodology. A lot of results lack methodology either.

<Author’s response>  

We appreciate your comments. Based on your feedback, we revised the chapter presenting the results in the same order as the methodology. We also revised the methodology part for more clarity. 

Moreover, in the results chapter, there are very often discussion elements, so it should either be separated or combined into one chapter the Results and discussion.

What has been called a discussion is a summary actually.

<Author’s response>

Thank you for your comment. We moved the discussion elements from the results chapter to the discussion chapter and edited discussion part entirely.

Pictures of O. glumaepatula, MER-IL113 and Kasalath are blurred.

<Author’s response>

We are sorry they are not clear which happened during online conversion. We changed the saving format of the pictures and present those in Fig. 2.

I am also not sure if all supplementary materials should be supplementary or be a part of the main article. Please, reconsider this once again.

<Author’s response>

Thank you for the comment. We moved the panels of RAE1 and RAE2 haplotypes from supplemental figure to Figure 5.

Summing up, the results presented in the paper are very good and should be published, but the form in which they were presented is unacceptable. Therefore, I propose to publish this manuscript after a thorough revision. I would like to emphasize that these are technical corrections - concerning the text, not merit. For the revision of this manuscript, please start by writing a comprehensive methodology. 

<Author’s response>

We appreciate your positive comment on our manuscript. In the revised version, we tried to make our manuscript legible and understandable.

Reviewer 3 Report

The manuscript was well written and the authors present results that I believe are novel and will be of interest to the scientific community. I have a few technical questions regarding the data analysis as well as some overall comments that I think would improve the quality and impact of the manuscript.

Questions/Comments:

Comment 1: The manuscript reads as two loosely related studies that could be better integrated 1) CSSL, and 2) SRA. The presentation of a new CSSL line provides excellent novelty to the manuscript, as does the identification of putative novel loci (I suggest that these novel loci be highlighted better in Figure 3 or 4). Then the manuscript delves into SRA analysis of other plant material, with a focus on known genes RAE1 and RAE2. It is unfortunate that additional effort was not made to use the SRA data to look at haplotypes or signatures of selection in other loci, particularly the novel loci from the CSSL lines, which might bring the two components of the study together more strongly.

Comment 2: The use of available data from SRA is an excellent use of community resources, however, I have some technical questions for the authors I am hoping they could please address.

 Can you please clarify on the ability to detect the TE in RAE1 in the methods. When you looked at ‘coverage’ do you mean presence of mapped reads across the region or do you mean ‘read depth’? In other words, was the filter that 20% of the region needed to have a read mapped to it or does the TE region need to have a read depth that is 20% that of the genome mean (or median) read depth? If it is having any data in the region, the analysis would be biased for samples with different levels of read depth. If you want to know if a region is present or absent, or has reduced read depth, there are ways to do so that have statistical support (i.e. bedtools + comparison to genome level depth, or tools such as CNVnator or cn.mop).

Along the same lines, is this TE (or a similar TE) present in other locations in the genome? If so, was there any effort to look at crossmapping of the sequencing reads to other locations? i.e. just because reads maps might not indicate that the TE is present if a similar TE is located elsewhere and the reads are crossmapping.

Does the O. meridionalis reference genome have the TE insertion? If not, would BWA allow reads from the insertion to map based on your alignment parameters? (i.e. I am wondering if the absence of the TE in the reference impacted your ability to detect the TE using BWA).

Perhaps I missed it, but how were the variants/haplotypes called (i.e. did you use a tool like freebayes, mpileup)?

Comment 3:

Perhaps it was an online conversion issue, but many of the figures have unusual formatting or poor placement of tracks/layers.

Minor Suggestions:

Line 21: “part of the domestication process of rice”…

Line 93: The number of datasets in SRA is irrelevant to this study if they are not related rice or even plants, consider removing this line. In addition, data outside of SRA is likely more pertinent to this study than some random microbe data in NCBI, i.e. ESTs from rice likely more relevant than a genome of bacteria.

Line 96: I find the wording confusing as to what is novel from this study and what is using material from other studies. Introducing the study in the middle of a large background paragraph also impacts the flow of ideas. Consider rewording and adding an additional line of context and perhaps start as a new paragraph that introduces the study. Perhaps as the start of the new paragraph: “In the present study, we evaluate 11 CSSLs for novel loci regulating awn development, which includes 1 CSSL that is novel to this study. The CSSL had a recurrent parent ….”

Line 111 and 117: 11 CSSL sets not 10? Or 10 sets “from previous studies”

RRESL I think is first used at line 117 but is not defined as to what it stands for.

Much of the results I would consider Materials and Methods, nearly the entire first paragraph describes the plant materials and methods used for testing. There are only a couple of lines that focus on results in section 2.1.

Figure 1B axis are illegible.

Figure 2 is poorly formatted and does not use the space well. Left most image in 2A could be enlarged, Figure 2 D has no vertical axis and the horizontal axis appears truncated. What do the error bars represent in Figure 2 D, please specify in caption.

Figure 3 is poorly presented. Instead of having an 11 panel figure, it could easily be presented in a way that’s more comparable to Figure 4. Or it could be a single panel figure with different colours representing the CSSL lines for each of the items being measured. Or perhaps 2-4 panels if they choose to present it separately for each parent. This way it could fit on a single page and be interpreted more easily. The figure is also poorly formatted, bars are overlapping, the chromosome boxes are not positioned well, some are different sizes and are missing labels. For example D appears to be chr. 1, 2,3,5,skips 6 and jumps to 7,blank?, 6, 7, 8 and etc.. Caption: grey squares, I only see black? What are even chromosomes? The axis are also not the same some are larger than others making it difficult to visually compare between lines.

Figure 4: Some features are overlapping i.e. Chr. 3 text around the centromere.

Line 230: “No previous study” is highly definitive/absolute. Perhaps “To our knowledge, this is the first study to report genes regulating awn development in these regions.”

Line 232: Missing indent of new paragraph?

Lines 233-234: “analyzing the functions” implies a ‘functional assay’ of some kind. Comparing sequences is not a functional assay. Perhaps “Compared sequences to identify mutations that might impact protein function”.

Line 241 and 389: I find “allele function” awkward. Perhaps “The loss of functional alleles of RAE1 and …”

Lined 250: using THE Sanger…

Were the sanger sequences of the genes deposited in NCBI? If not, I think they are in the alignments in supplemental which would be fine.

Line 269 seems like an incomplete thought, is the logic that these chromosome should carry RAE1and RAE2?

Line 256: Consider adding this as a footnote to Table 2 because if looking at table 2 on its own it would make it appear that it should be awned based on RAE1 and RAE2.

Line 288: reword suggestion: Sequencing reads from 486 individuals were mapped to the reference genome of O. sativa ssp. Japonica and SNPs called.

What are the blue text in table S4?

Line 435: ‘may be’ essential – additional information/experiments I think are required to show that they ‘are’ essential.

Line 450: definitive – is it possible that other architectures may be better in the future or is this the true optimum?

I thought the discussion was long and in some sections read like a literature review and the link to the study and results is unclear. For example, second last paragraph of discussion.

Author Response

Comments and Suggestions for Authors

The manuscript was well written and the authors present results that I believe are novel and will be of interest to the scientific community. I have a few technical questions regarding the data analysis as well as some overall comments that I think would improve the quality and impact of the manuscript.

<Author’s response>

Thank you so much for reviewing our manuscript and for giving us questions, comments and suggestions. We modified and improved the manuscript point by point according to the reviewer’s comments.

Questions/Comments:

Comment 1: The manuscript reads as two loosely related studies that could be better integrated 1) CSSL, and 2) SRA. The presentation of a new CSSL line provides excellent novelty to the manuscript, as does the identification of putative novel loci (I suggest that these novel loci be highlighted better in Figure 3 or 4). Then the manuscript delves into SRA analysis of other plant material, with a focus on known genes RAE1 and RAE2. It is unfortunate that additional effort was not made to use the SRA data to look at haplotypes or signatures of selection in other loci, particularly the novel loci from the CSSL lines, which might bring the two components of the study together more strongly.

<Author’s response>

Thank you for your comments to make our manuscript better. The RAE1 and RAE2 have been identified in our previous studies, giving us the information of their exact locations and the sequences near the 2 loci. This made us possible to do haplotype analysis using SRA. On the other hand, other loci have not been identified yet. In the current status, the candidate chromosome regions of other loci are still huge. There are enormous polymorphisms in the region among species, making it difficult to perform sequence comparison. In the near future, we will confine the candidate chromosome region of unidentified genes by mapping strategy. We will try to pursue haplotypes analysis or signatures of the selection of these loci using SRA data.

Comment 2: The use of available data from SRA is an excellent use of community resources, however, I have some technical questions for the authors I am hoping they could please address.

 Can you please clarify on the ability to detect the TE in RAE1 in the methods. When you looked at ‘coverage’ do you mean presence of mapped reads across the region or do you mean ‘read depth’? In other words, was the filter that 20% of the region needed to have a read mapped to it or does the TE region need to have a read depth that is 20% that of the genome mean (or median) read depth? If it is having any data in the region, the analysis would be biased for samples with different levels of read depth. If you want to know if a region is present or absent, or has reduced read depth, there are ways to do so that have statistical support (i.e. bedtools + comparison to genome level depth, or tools such as CNVnator or cn.mop).

<Author’s response>

Thank you for your comments. We apologize for the confusion caused by our inaccurate wording. The coverage means the breadth of sequence alignments estimated by bedtools coverage, and we did not see the read depth. The filter was that the reads needed to be mapped to 80% of the target region’s breadth (both TE boundaries; inside of TE (200 bp) and outside of TE (200 bp)). We concluded that there is the TE in RAE1 promoter region if the reads covered 80% breadth of the target region. We clarified this part in the methodology part.

Along the same lines, is this TE (or a similar TE) present in other locations in the genome? If so, was there any effort to look at crossmapping of the sequencing reads to other locations? i.e. just because reads maps might not indicate that the TE is present if a similar TE is located elsewhere and the reads are crossmapping.

<Author’s response>

We appreciate your comments. Multiple same sequence of the TEs (MuDR) are located in O. sativa japonica genome. Thus, we focused only on both sides of the TE insertion boundary. There were no reads mapped to the internal regions of the TEs because the multi-mapped reads into the rice genome were excluded from the analysis. On the other hand, if one read of a paired-end read was uniquely mapped to the edge of TE (200 bp) or the outside sequence of the RAE1 TE region (200 bp) which is specific to target location on chromosome 4, we interpreted that there is the TE insertion in RAE1 promoter region. We clarified this part in the methodology and result part.

Does the O. meridionalis reference genome have the TE insertion? If not, would BWA allow reads from the insertion to map based on your alignment parameters? (i.e. I am wondering if the absence of the TE in the reference impacted your ability to detect the TE using BWA).

Perhaps I missed it, but how were the variants/haplotypes called (i.e. did you use a tool like freebayes, mpileup)?

<Author’s response>

Thank you for your comments. Short reads of O. meridionalis were not mapped around TE insertion area in O. sativa japonica because of small many indels even we changed the BWA parameters. O. meridionalis reference genome does not have the TE insertion. We confirmed that the short reads from O. meridionalis were mapped seamlessly on the promoter region of O.meridionalis RAE1 where is the orthologous region of the TE insertion site in O.sativa japonica (Supplemental Figure 6B). We clarified this part in the methodology and added Supplemental Figure 6B. We did the variants/haplotypes call by mpileup.

Comment 3:

Perhaps it was an online conversion issue, but many of the figures have unusual formatting or poor placement of tracks/layers.

<Author’s response>

Thank you for the comment. Some figures showed unusual formatting because of the online conversion. We changed the saving format to address this issue.

Minor Suggestions:

Line 21: “part of the domestication process of rice”…

<Author’s response>

Thank you for pointing this out. We changed the sentence according to the reviewer’s suggestion.

Line 93: The number of datasets in SRA is irrelevant to this study if they are not related rice or even plants, consider removing this line. In addition, data outside of SRA is likely more pertinent to this study than some random microbe data in NCBI, i.e. ESTs from rice likely more relevant than a genome of bacteria.

<Author’s response>

We appreciate your suggestion. We removed the sentence about the numbers of datasets in SRA.

Line 96: I find the wording confusing as to what is novel from this study and what is using material from other studies. Introducing the study in the middle of a large background paragraph also impacts the flow of ideas. Consider rewording and adding an additional line of context and perhaps start as a new paragraph that introduces the study. Perhaps as the start of the new paragraph: “In the present study, we evaluate 11 CSSLs for novel loci regulating awn development, which includes 1 CSSL that is novel to this study. The CSSL had a recurrent parent ….”

<Author’s response>

Thank you for the comment. We changed the sentence based on reviewer’s suggestion.

Line 111 and 117: 11 CSSL sets not 10? Or 10 sets “from previous studies”

<Author’s response>

We added the word “from previous studies” after 10 sets of CSSL.

RRESL I think is first used at line 117 but is not defined as to what it stands for.

<Author’s response>

Thank you for pointing this out. RRESL is the name of CSSL we developed, not abbreviation. We modified the sentence as “we developed a new CSSL named RRESL whose donor parent is O. glumaepatula…”.

Much of the results I would consider Materials and Methods, nearly the entire first paragraph describes the plant materials and methods used for testing. There are only a couple of lines that focus on results in section 2.1.

<Author’s response>

We rewrote the section 2.1 and some paragraphs were moved to materials and methods part.

Figure 1B axis are illegible.

Figure 2 is poorly formatted and does not use the space well. Left most image in 2A could be enlarged, Figure 2 D has no vertical axis and the horizontal axis appears truncated. What do the error bars represent in Figure 2 D, please specify in caption.

<Author’s response>

Thank you for pointing this out. We changed the Fig. 1B showing a clearer label. We scaled the left image in Fig. 2A the same with the other two images and changed the alignment of the panels. We added the vertical axis title, fixed the label of the horizontal axis, and added the information of error bars of Fig. 2D in figure legend.

Figure 3 is poorly presented. Instead of having an 11 panel figure, it could easily be presented in a way that’s more comparable to Figure 4. Or it could be a single panel figure with different colours representing the CSSL lines for each of the items being measured. Or perhaps 2-4 panels if they choose to present it separately for each parent. This way it could fit on a single page and be interpreted more easily. The figure is also poorly formatted, bars are overlapping, the chromosome boxes are not positioned well, some are different sizes and are missing labels. For example D appears to be chr. 1, 2,3,5,skips 6 and jumps to 7,blank?, 6, 7, 8 and etc.. Caption: grey squares, I only see black? What are even chromosomes? The axis are also not the same some are larger than others making it difficult to visually compare between lines.

<Author’s response>

We apologize for the quality of Fig.3. Some errors were due to online conversion. We selected few panels instead of showing all 11 panels and the remaining panels were moved to the supplemental figure to fit Fig. 3 into one page. We reformatted the bar graphs without overlapping, missing labels are added, and the y-axis values are put same number depends on recurrent parent (2 cm for T65 background CSSLs, 6 cm for Koshihikari background CSSLs). In previous version, Mer-IL[Mer] has missing chromosome number, this is because 2 lines (Mer-IL[Mer] 121 and 122) has no chromosome substitution. So, we removed these 2 lines and aligned individual lines along with the chromosome number harboring by donor parent chromosome in Mer-IL[Mer]. In the end, the line number of Mer-IL[Mer] is ordered from 101 to 118, then 123,124, 119, 120, then 125 to 136 (Supplemental Figure 2B).

Figure 4: Some features are overlapping i.e. Chr. 3 text around the centromere.

<Author’s response>

Thank you for your comment. We improved the Fig. 4 for clarity.

Line 230: “No previous study” is highly definitive/absolute. Perhaps “To our knowledge, this is the first study to report genes regulating awn development in these regions.”

<Author’s response>

Thank you for your suggestion. We changed the sentence based on reviewer’s suggestion.

Line 232: Missing indent of new paragraph?

<Author’s response>

We added the indention.

Lines 233-234: “analyzing the functions” implies a ‘functional assay’ of some kind. Comparing sequences is not a functional assay. Perhaps “Compared sequences to identify mutations that might impact protein function”.

<Author’s response>

Thank you for the correction. We changed the sentence according to the reviewer’s suggestion.

Line 241 and 389: I find “allele function” awkward. Perhaps “The loss of functional alleles of RAE1 and …”

Lined 250: using THE Sanger…

<Author’s response>

Thank you for the correction. We changed the sentence according to the reviewer’s suggestion.

Were the sanger sequences of the genes deposited in NCBI? If not, I think they are in the alignments in supplemental which would be fine.

<Author’s response>

Thank you for suggestion. We added the sequence alignment of RAE1 and RAE2 in Supplemental Fig. 2.

Line 269 seems like an incomplete thought, is the logic that these chromosomes should carry RAE1and RAE2?

<Author’s response>

We apologize for the confusion. The sanger sequencing results showed that W1625 carries functional RAE1 and RAE2. Hence, we omitted this sentence.

Line 256: Consider adding this as a footnote to Table 2 because if looking at table 2 on its own it would make it appear that it should be awned based on RAE1 and RAE2.

<Author’s response>

We added that the awnlessness in O. glaberrima was caused by loss of function of RAE3 through African rice domestication, not by RAE1 and RAE2 selection.

Line 288: reword suggestion: Sequencing reads from 486 individuals were mapped to the reference genome of O. sativa ssp. Japonica and SNPs called.

<Author’s response>

Thank you for your suggestion. We changed the sentence based on reviewer’s suggestion.

What are the blue text in table S4?

<Author’s response>

We are sorry for this mistake. We changed the blue color to black.

Line 435: ‘may be’ essential – additional information/experiments I think are required to show that they ‘are’ essential.

<Author’s response>

Thank you for the instruction. We added “may be” in this sentence.

Line 450: definitive – is it possible that other architectures may be better in the future or is this the true optimum?

I thought the discussion was long and in some sections read like a literature review and the link to the study and results is unclear. For example, second last paragraph of discussion.

<Author’s response>

Thank you for your important comments. We rewrote the discussion part thoroughly to be shorter and not to be like a literature review. Mainly, we omit the part about the improvement of current cultivar by manipulating awn-related genes.

Reviewer 4 Report

The authors generated a novel CSSL population for O. glumaepatulata for analysis of loci associated with awn length.  This was combined with CSSL populations for 10 other wild rice species.  This analysis confirmed the importance of RAE1 and RAE2, especially in asian rice domestication.  Haplotype analysis of these genes indicates that multiple alleles of these key genes were selected.  It was also reconfirmed that African rice is awnless due to a seperate gene (RAE3). 

I found minor errors in the text and figures that need to be corected before publication.

  • Line 18 - remove the word inhbition
  • line 20 - should read "modern rice agriculture."
  • Figure 2 D - Small marks are present betwen the chart and labels
  • Figure 3 (Line 187) - Where are the grey Squares?  
  • Figure 3 - Indicate in the legend what the numbers above the Chr. boxes represent.
  • Line 366 - should read "function had a big impact...)
  • Line 403 - Should read " glaberrima was not caused by alteration of RAE1 or RAE2"
  • Line 420 - remove the word on
  • Line 450 - should be "reached a close to optimal plant actchitecture...."
  • Line 465 - It is not clear what you mean by through optimal gene and genetic backgrounds. 

Author Response

Comments and Suggestions for Authors

The authors generated a novel CSSL population for O. glumaepatulata for analysis of loci associated with awn length.  This was combined with CSSL populations for 10 other wild rice species.  This analysis confirmed the importance of RAE1 and RAE2, especially in asian rice domestication.  Haplotype analysis of these genes indicates that multiple alleles of these key genes were selected.  It was also reconfirmed that African rice is awnless due to a seperate gene (RAE3). 

I found minor errors in the text and figures that need to be corected before publication.

  • Line 18 - remove the word inhibition
  • line 20 - should read "modern rice agriculture."
  • Figure 2 D - Small marks are present between the chart and labels
  • Figure 3 (Line 187) - Where are the grey Squares?  
  • Figure 3 - Indicate in the legend what the numbers above the Chr. boxes represent.
  • Line 366 - should read "function had a big impact...)
  • Line 403 - Should read " glaberrima was not caused by alteration of RAE1 or RAE2"
  • Line 420 - remove the word on
  • Line 450 - should be "reached a close to optimal plant actchitecture...."
  • Line 465 - It is not clear what you mean by through optimal gene and genetic backgrounds. 

<Author’s response>

We appreciate your review of our manuscript and indicating the mistake. We changed the words and sentences as based on reviewer’s suggestion. Fig.3 was changed for clarity, and we added the explanation about the numbers above the chromosome box. Those numbers are line number of each CSSL. Gray squares might disappear after online conversion, hence, we changed the saving format to show gray squares. In discussion part, we omit the part about the improvement of present cultivar by manipulating awn-related genes including line 450 and 465.

Round 2

Reviewer 2 Report

Dear Authors,

I really like the current version of manuscript.

Lines 308-311: (123 wild and 52 cultivated rice; 18 O. barthii, 19 308 O. glaberrima,  etc.) - Maybe would be better (52 cultivated rice and 123 wild including: 18 O. barthii, 19 308 O. glaberrima etc.)

Line 564: Should be added that mpileup is a part of Samtools.

Best regards,